# Keep Your Eggs Away: Ant Presence Reduces *Ceratitis capitata* Oviposition Behaviour through Trait-Mediated Indirect Interactions

**DOI:** 10.3390/insects14060532

**Published:** 2023-06-07

**Authors:** Stefania Smargiassi, Alberto Masoni, Filippo Frizzi, Paride Balzani, Elisa Desiato, Giovanni Benelli, Angelo Canale, Giacomo Santini

**Affiliations:** 1Department of Biology, University of Florence, 50121 Firenze, Italy; stefania.smargiassi@unifi.it (S.S.); alberto.masoni@unifi.it (A.M.); elisa.desiato@stud.unifi.it (E.D.); giacomo.santini@unifi.it (G.S.); 2National Biodiversity Future Center, 90133 Palermo, Italy; 3South Bohemian Research Center of Aquaculture and Biodiversity of Hydrocenoses, Faculty of Fisheries and Protection of Waters, University of South Bohemia in Ceske Budejovice, Zátiší 728/II, 389 25 Vodňany, Czech Republic; 4Department of Agriculture, Food and Environment, University of Pisa, 56124 Pisa, Italy; giovanni.benelli@unipi.it (G.B.); angelo.canale@unipi.it (A.C.)

**Keywords:** Mediterranean acrobat ant, *Tapinoma nigerrimum* complex, ant semiochemicals, insect fruit pest, medfly

## Abstract

**Simple Summary:**

The use of chemical pesticides in agriculture is a critical threat to the environment. Implementing the use of biological control practices is an increasing worldwide challenge to cope with this matter. The exploitation of trait-mediated indirect interactions (TMIIs), which is an avoidance behaviour of pests when detecting possible risk, is a new and interesting pathway to follow. Ants, which are predators of many insect pests, are commonly active on plants and release several different chemical traces in the substrate, making them potential candidates for TMII-based management approaches. We tested whether semiochemicals released by two Mediterranean ants, *Crematogaster scutellaris* and *Tapinoma nigerrimum*, are able to deter the occurrence of a strongly harmful pest of tree crops, the tephritid *Ceratitis capitata*, which lays eggs within fruits. Using binary choice tests between a plum previously visited by ants and another used as control, we actually observed an avoidance behaviour by females of *C. capitata*, which results in a lower amount of progeny production, suggesting that flies can detect the chemical compounds released by ants. This study suggests that scents triggering this deterrence effect are conserved across ant subfamilies and encourages improving this research to achieve a new low-impacting control method against agricultural pests.

**Abstract:**

Alternative methods to achieve sustainable agricultural production while reducing the use of chemical pesticides, such as biological control, are increasingly needed. The exploitation of trait-mediated indirect interactions (TMIIs), in which pests modify their behavior in response to some cues (e.g., pheromones and other semiochemicals) to avoid predation risk, may be a possible strategy. In this study, we tested the effect of TMIIs of two Mediterranean ant species, *Crematogaster scutellaris* and *Tapinoma nigerrimum*, on the oviposition behaviour of *Ceratitis capitata* (Diptera: Tephritidae), one of the world’s most economically damaging pests, which attacks fruits. For each ant species, we performed choice experiments using ant-scented and control plums, counting the time spent by medflies on fruits and the number of pupae emerging from them. Results of both ant species tests showed a significantly shorter time spent by ovipositing medflies on ant-exposed plums and a lower number of pupae, when compared to the control group. Our findings highlighted that the semiochemicals released by ants on plums triggered an avoidance behaviour by medfly females, leading to lower oviposition rates. This study contributes to the understanding of indirect ant–pest interactions in Mediterranean agricultural settings and points out the potential of utilising ant-borne semiochemicals in sustainable IPM strategies.

## 1. Introduction

The control of insect pests in agriculture is usually achieved using chemical insecticides, but their widespread use has resulted in increased environmental pollution and rising costs [1] The increasing awareness of the many environmental problems generated by the use of agrochemicals has encouraged the development of alternative methods to make agricultural production ecologically and economically sustainable [2,3,4], thus preserving environmental quality and maintaining high biodiversity levels [5].

A way to reduce the use of agrochemicals is to rely on biological control. This method allows for controlling pest populations by exploiting the natural relationship of antagonism or predation among species [6,7,8]. The control of pest populations through direct consumptive interactions and reduction in pest abundance is a well-established field of research that gained attention in the last decades [9]. The control of a pest population can also be achieved through non-consumptive interactions. This is the case when the pest species can perceive the predation risk, for example, when sensing the presence of the predator, and it can modify its behaviour in order to escape it. In this way, the effects of the pest are locally and temporarily reduced, with important cascade effects on the whole community [10,11,12]. These interactions are called trait-mediated indirect interactions (TMIIs) because they are due to the modification of a prey’s trait, rather than a change in its density [13]. One of the most interesting and useful aspects of TMIIs is that these can occur even in the absence of the predator, as long as their indirect traces are present in the environment [11,14,15]. Chemical traces, such as kairomones (compounds provoking favourable adaptive changes in the behaviour of a receiver but not in the giver [16]), are examples of warning cues of its recent passage for a potential prey [17].

Among insect taxa potentially useful in biological control, ants deserve a special role, as several empirical data suggest that the use of ants in integrated pest management can promote sustainable yields in agricultural systems [6,8,18]. Colonial recruitment behaviour, which allows the rapid occurrence of many individuals on a potential resource, could result in massive attacks of insect pests and undermine predator satiation-based defensive strategies [19]. Moreover, ants produce many different types of pheromones used for intra- and inter-specific communication, as well as during foraging and patrolling activities [20,21,22,23], which might be detected by the pest [17]. This makes this taxon a good candidate for biological control through TMIIs. One of the most striking examples is provided by the weaver ant *Oecophylla smaragdina* (Fabricius, 1775) and *O. longinoda* (Latreille, 1882), the presence of whom are currently known to control more than 50 different arthropod pests in 12 different woody crops in tropical areas, both by direct and indirect interactions with pests [24,25,26]. In some cases, this control has led to an increase of more than 70% in farmers’ net income by replacing the conventional pesticide [1]. Van Mele et al. [27] found that the laying behaviour of the two important mango pests, the fruit flies *Ceratitis cosyra* (Walker, 1849) and *Bactrocera invadens* (Drew-Tsuruta and White, 2005), was influenced by the pheromones of the dominant ant *O. longinoda*. Flies were unwilling to land on fruits treated with ant secretions and, once landed, often left quickly without laying eggs. Field experiments have shown that the closer the fruit is to the ant nest, the lower the damage of the flies [26]. It was, therefore, hypothesized that *O. longinoda* releases pheromones that influence fly behaviour, even when the ant is not physically present [26]. Most of the available reports, however, come from tropical habitats, while little is known about indirect ant–pest interactions in temperate agroecosystems [17,26,27].

The Mediterranean fruit fly *Ceratitis capitata* (Wiedemann, 1824), is one of the most important crop pests worldwide [28]. This polyphagous insect attacks a wide range of host species that include more than 350 fruits and vegetables [29]. The species is native to sub-Saharan Africa, but it can be found in many tropical, subtropical, and temperate regions worldwide due to its ability to adapt to different climates, as well as its high invasive potential [30,31]. Traditionally, the control of this dipteran is achieved using organophosphates and pyrethroid insecticides [32]. Other approaches, such as mass trapping, the sterile insect technique, and biological control, are possible options, although they are not as of yet utilised [33].

In this study, we investigated the potential role of two common ant species in the indirect control of *C. capitata*. The studied species were the acrobat ant *Crematogaster scutellaris* (Olivier, 1792) and a species belonging to the *Tapinoma nigerrimum* complex (see [34] for a complete revision of the taxonomy of this group), hereafter referred to as *Tapinoma nigerrimum* (Nylander, 1856) for simplicity. Both species are dominant and widespread throughout the western Mediterranean basin, occupying both natural and human-managed ecosystems [35,36,37,38]. The large monogynous colonies of *C. scutellaris*, reaching numbers of several thousand individuals [39] and their nesting habits in trees [40,41], make this species particularly suitable as a model species for testing the role of ants in biological control of woody crops, being a highly efficient predator also in tree canopies [42,43,44,45]. *Tapinoma nigerrimum* nests in the soil, but it regularly visits nearby trees to tend aphids and mealybugs, even in disturbed contexts [46]. Since the two species belong to two different subfamilies (Myrmicinae and Dolichoderinae, respectively), our aim was also to investigate whether there was a difference between the possible repellent effect of the semiochemicals produced by the two species.

We performed multiple choice experiments to test whether the chemical traces laid by workers of the two ant species on fruits (plums, *Prunus domestica* L, 1753) deter mature females of *C. capitata*, reducing its oviposition behaviour. In particular, we evaluated (i) whether the oviposition rate on ant-scented and control plums were different, and (ii) whether the time spent by flies to oviposit on ant-scented and control plums differed. According to previous studies on *Oecophylla* spp., we hypothesized a lower preference and a lower oviposition rate in treated fruits due to a shorter period spent in ant-scented plums [26].

## 2. Materials and Methods

The experiments were conducted in July 2021 (*C. scutellaris*) and 2022 (*T. nigerrimum*) at the Department of Biology of the University of Florence in the Sesto Fiorentino Science Campus (43°49′03.5″ N 11°12′14.3″ E). About 1200–1500 adult workers of the two ant species were collected near the campus. Ants were housed in separate plastic containers (15 cm × 23 cm × 15 cm, about 600–700 workers each) with Fluon-coated walls (AGC Chemicals Europe, Ltd., York House, Thornton-Cleveleys FY5 4QD, UK) to avoid ants escaping.

*Ceratitis capitata* pupae, close to eclosion, were provided by the Entomology Section of the University of Pisa, where this species is routinely reared following the protocol described by Benelli et al. [47]. The pupae were placed inside a thermostatic chamber with constant temperature (25 °C), and humidity (relative humidity = 80%), and exposed to natural illumination until eclosion. Food (brewer’s yeast, powdered sugar) and water were provided ad libitum. Each day, the chambers were checked to collect newly eclosed flies, which were then moved by an insect aspirator to new flight chambers to form equally aged cohorts of flies. Adult flies were maintained under the same conditions for an additional 10 days prior to the start of the experimental trials to allow sexual maturation and mating to occur [48].

Several factors can influence the laying behaviour of Mediterranean fruit flies, such as fruit type, size, quality, or the presence of chemicals [49,50]. We used organic plums (cv. Stanley) purchased in a local market, from the same production batch and having similar sizes and degrees of ripeness. Flies had no previous experience of the fruits, being born in captivity. Before being used in the experiments, all plums were washed with tap water and blotted with absorbent paper to remove possible chemical residuals from the exocarp. Plums and laboratory equipment were handled while wearing sterile nitrile gloves to prevent the transfer of chemical compounds from the hands to their surface. For each experiment, we randomly selected 40 plums, which were then divided into two groups for subsequent experimental tests (20 for ant treatment and 20 as controls). Ant-treated plumes were placed inside a box containing approximately 600 freshly collected workers of one of the two species. Ants were supplied only with water ad Libitum. Control plums were housed in exactly the same conditions, but without ants. In both groups, adhesive paste (UHU^®^Patafix) was put around the stem of the plums to prevent ants from piercing them. Plums were kept under natural light and room temperature for 48 h, which is the time required for the pheromones released by the ants to settle on the fruit [27,51].

### 2.1. Experimental Setup

To assess the preference of *C. capitata* for treated or untreated fruits, we used cylindrical Polyethylene Terephthalate (PET) experimental chambers (diameter: 25 cm, length: 70 cm), having a 10 cm inlet opening on the centre-top part. Both ends of the chamber and the inlet opening were made of transparent chiffon fabric to ensure air flux and avoid saturation of the volatiles (Figure 1). Twenty experimental binary choice tests were set up for each ant species.

During each test, two plums, one treated and one control, were placed at the opposite ends of the chamber (Figure 1). Treated plums were placed immediately after ants were gently removed from their surface with a soft drawing brush. The position of control and treated plums were alternated in the different chambers so as to randomize uncontrolled effects. Three *C. capitata* females (10 days old) were introduced in the center of the chamber through the top inlet and left inside the chamber for three consecutive days to oviposit.

After this period, plums were removed and placed individually inside sterile glass jars. The jars had 1 cm of moist sand (construction sand, 0–1 mm size, Gras Calce S.r.l.) on the bottom to allow the eclosing larvae to metamorphose into pupae and were covered with a fine mesh to prevent their escape. The jars were housed at 25 °C and with 80% relative humidity for 4 weeks, and, once a week, we counted the number of pupae found sifting the sand [27].

In parallel with the oviposition choice test, we carried out direct observations of the flies’ oviposition behaviour. Each day, we observed the chambers four times (at 10:00 A.M., 12:00 P.M., 3:00 P.M., and 5:00 P.M.) for 10 min, and we measured the total time spent by the flies on the plums.

### 2.2. Statistical Analysis

Total time spent was analysed using linear mixed-effects models (LMMs), with treatment and days (1 to 3) as fixed factors and chamber ID as a random term. The number of pupae was analysed using Mixed-effects Generalised Linear Models (GLMMs) with a Poisson distribution, using treatment (ant-scented/control) as a fixed factor and test chamber ID as a random term. All analyses were performed in R, v4.0.4 [52], using the packages “lme4” [53], and “lmerTest” [54]. The significance threshold for all tests was set at *p* = 0.05.

## 3. Results

The time spent by flies after landing on control plum fruits was significantly longer than on treated plums in both *C. scutellaris* (*F_1,95_* = 7.030, *p* < 0.01) and *T. nigerrimum* (*F_1,95_* = 6.311, *p* < 0.05) tests (Figure 2a,b, respectively). Only in *C. scutellaris* tests, the time spent changed also over the three days (*F_2,95_* = 3.417, *p* < 0.05, Figure 2a).

The number of pupae retrieved from treated plums was significantly lower than in control plums, both when treated with *C. scutellaris* (GLMM, *z* = −8.641, *p* < 0.001) and *T. nigerrimum* (GLMM, *z* = −7.921, *p* < 0.001) (Figure 3). From plums treated with *C. scutellaris* (Figure 3a), the average number of pupae retrieved was 8.55 (±2.07 SE), compared to 18.85 (±3.47 SE) from controls. Whereas, from plums treated with *T. nigerrimum* (Figure 3b), the average number of pupae was 1.3 (±0.7 SE) compared to 7.5 (±2.19 SE) from controls.

## 4. Discussion

The results of this study provide new insights into the TMIIs of ants on the oviposition behaviour of the Mediterranean fruit fly *C. capitata*. We showed that *C. capitata* preferred to oviposit on fruits not previously visited by ants, a result highlighted by both the number of pupae and the behaviour of flies, which tend to avoid staying on treated fruits. This evidence suggests that ants deposit one or more persistent compounds on fruits that would deter *C. capitata* from visiting the plums.

Although the chemical nature of these substances is not yet known, the fact that deterrence was triggered by both species suggests that the compounds involved in these interactions—which could be assumed to be kairomones, considering *Oecophylla* sp. as a model [55]—might be common to the different ant subfamilies (Myrmicinae and Dolichoderinae), despite their considerable taxonomic distance. Moreover, the similar effects described in *Oecophylla* [26,27,56], which belongs to the Formicinae subfamily, further supports this idea. Among the substances involved in chemical communication in ants, and candidates for being the kairomones triggering the observed effects, cuticular hydrocarbons (CHCs) might play a special role. CHCs are pheromones commonly used by ants and other insects to recognize nestmates and/or mutualists, but they may also be released in the environment and be involved, for example, in marking the home range [57,58], and they can be also used as kairomones [59]. Unfortunately, despite the fact that the CHCs of the two studied species have been analysed in previous studies (e.g., [22,35] for *C. scutellaris*, [60] for the *T. nigerrimum* complex), a comparative analysis to identify the effective molecules on available data is difficult because of the different laboratory standards used. Of note, it is very well known that the CHC profile of a colony can change very rapidly, even under the effect of very local factors, such as food sources used [35], but our results suggest that the substances triggering the effect are quite stable, as they are conserved across subfamilies.

In addition to CHCs, other pheromones are commonly released by ants in the substrate during patrolling or foraging, for example, trail pheromones [61]. This family of pheromones includes many different compounds, and workers sometimes release a mix of them, having different characteristics of persistence and communication meaning [62]. Hence, defining a precise chemical compound that is responsible for the observed effects is not simple, and, also, in this case, a targeted comparative analysis is needed. One important factor to consider is that the pheromones detected by flies should not be extremely volatile, since the effect persists even when ants are not present, and we recorded an observable effect—though reduced—even three days after the treatment. Pheromone persistence may depend on external factors, such as temperature, which may lead to a higher rate of evaporation [63], or the capacity of the substrate to hold the compound [64]. Beugnon and Dejean [65] reported that trail pheromones released by *O. longinoda* were still identifiable after nine weeks, and anal spots persisted for up to ten months, even if they had been washed by rain. *Tapinoma simrothi* (Emery, 1925) workers use persistent tracks to seek aphid honeydew sources, a feature helping expand the foraging network [66]. On the other hand, trail pheromones of the garden ant *Lasius niger* (Linnaeus, 1758) can persist for only 20–24 h without a continuous passage of workers [67], but, in other species, some pheromones may be even less durable (e.g., 30 min); this is an adaptive trait for species that exploit ephemeral food sources because it facilitates the rapid abandonment of depleted sources, as occurs in *Linepithema humile* (Mayr, 1868) and *Monomorium pharaonis* (Linnaeus, 1758) [64,68]. Here, short duration pheromones should be excluded. It is finally possible that flies merely habituate to the ants’ odour, thus the avoidance effect decreased with time; however, previous studies would suggest that this species does not habituate to odours [69]. Further ad hoc experiments will need to investigate this point.

From a practical point of view, our results suggest that the presence of ants on trees might be, overall, useful to protect crops from the Mediterranean fruit fly. In fact, active ant colonies continuously release pheromones on the substrate, at least during the activity period, usually between March and October, in the Mediterranean basin, depending on the species and local climate [61]. Predation, in particular of juvenile stadia, could be an additional benefit of the presence of ants. Previous studies have shown that *C. capitata* can be directly preyed on by several ant species [70,71,72]. *Crematogaster scutellaris* has been observed to predate larvae of other dipterans on olive trees [44], although the preys were not fruit flies. Moreover, *C. scutellaris* naturally nests in tree trunks, dead logs, or wooden structures, and all these nesting sites may be available in cultivated fields [35,40]. Its polydomic colony organisation, with a colony subdivided into several satellite interconnected nests, make this species potentially highly diffuse in tree crops [42]. Similarly, *T. nigerrimum* is widespread in different agrosystems, such as olive groves [73,74], vineyards, and citrus groves [46]. Although the nests of *T. nigerrimum* are dug in the ground, workers can visit nearby trees and plants to predate other arthropods and/or to exploit hemipterans’ honeydew [75].

In this sense, it should be emphasized that ants can be not only a friend for plants, but even a foe, since they are tenders of other kind of pests, such as aphids or scales, that usually increase in number when co-occurring with ants [74,76]. *Crematogaster scutellaris* and *T. nigerrimum*, from this perspective, are not an exception. Hence, the occurrence of ants can be a double-edged sword in protecting crops.

Alternatively, once the substances potentially involved are identified, and once the possibility of industrial production of them has been evaluted, they might be used as a natural and low-impacting deterrent against *C. capitata*, which could provide beneficial effects to plants without using non-target chemical pesticides. However, as for all other chemical compounds used in agriculture, many parameters should be tested before the compound is employed, such as volatilization, runoff, drift, leaching, degradation, soil sorption, and several others, which are all needed to verify the efficiency and the impact on the local organisms of the compound [77]. Most of these parameters depend on the surrounding environment, and an in-depth analysis of the agroecosystem is a concern unavoidable before using chemicals, particularly when they are new. Finally, although natural-based, the effects of such substances on the communities of both animals and plants should not be underestimated, and detailed analyses are strongly recommended.

## 5. Conclusions

To our knowledge, this study is the first one reporting TMIIs between ants and a diffuse highly harmful pest, such as *C. capitata* in the Mediterranean basin. Results may open new, applied avenues for biological control based on TMIIs. Natural ant semiochemicals could provide great benefits to plants, always keeping in mind the possible double effect that the presence of ants might have for a crop. However, if the compounds that are responsible for this avoidance effect will be identified, isolated, and eventually synthesized for industrial production, and then they will be demonstrated to have lower impacts on the environment with respect to other pesticides, this might be a very important step ahead toward the biological fight against one of the most striking pests over the Mediterranean fruit orchards.

## Figures and Tables

**Figure 1 insects-14-00532-f001:**
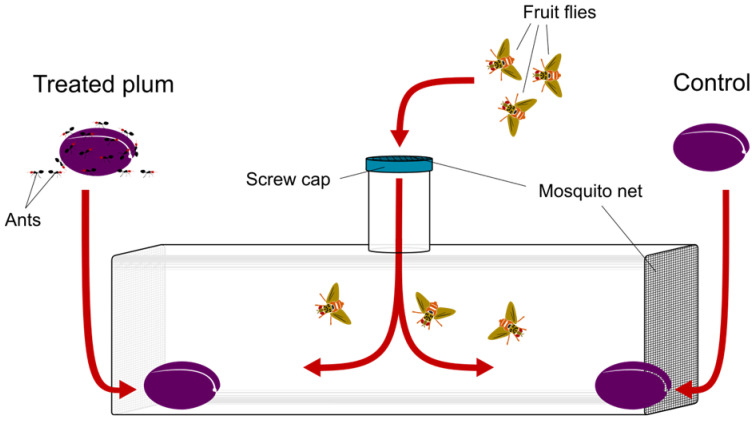
Experimental apparatus and set up of the two-choice trials.

**Figure 2 insects-14-00532-f002:**
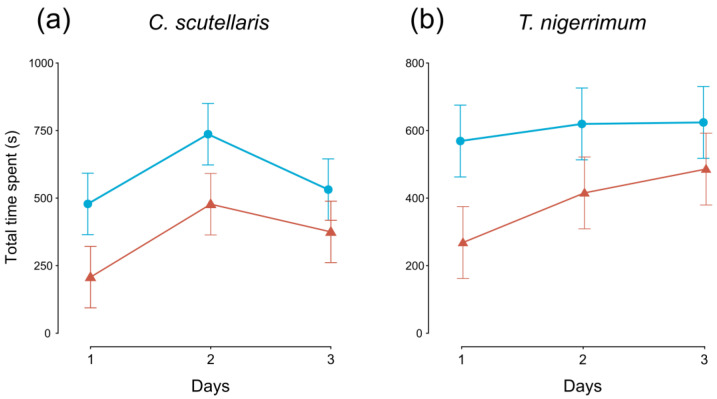
Time (average ± SE) spent by *Ceratitis capitata* flies on untreated (control, blue lines, and circles) and treated (red lines and triangles) plums over the three days. Plums treated with (**a**) *Crematogaster scutellaris* or (**b**) *Tapinoma nigerrimum* scents.

**Figure 3 insects-14-00532-f003:**
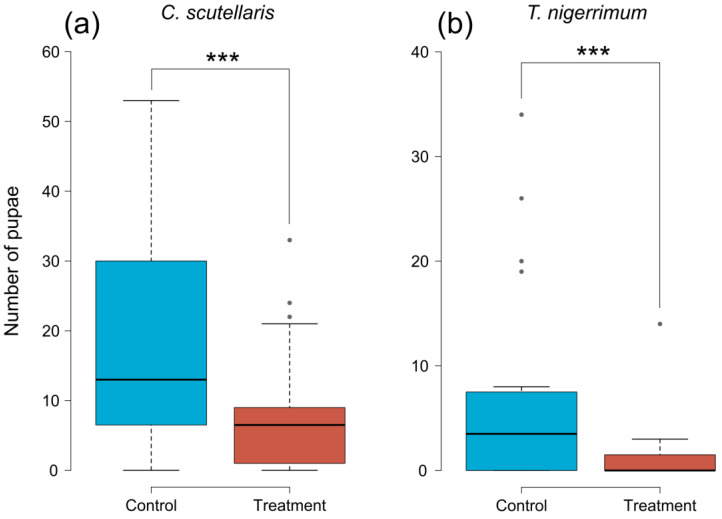
Number of pupae of *Ceratitis capitata* retrieved from untreated (controls, blue) and treated (red) plums. Plums treated with (**a**) *Crematogaster scutellaris* or (**b**) *Tapinoma nigerrimum* scents. The dots and dashed lines represent the differences between different treatments. Asterisks represent the significance level (*** = *p* < 0.001).

## Data Availability

Smargiassi, Stefania; Masoni, Alberto; Frizzi, Filippo; Balzani, Paride; Desiato, Elisa; Benelli, Giovanni; et al. (2023). Dataset of “Keep your eggs away: ant presence reduces *Ceratitis capitata* oviposition behaviour through trait-mediated indirect interactions”. figshare. Dataset. https://doi.org/10.6084/m9.figshare.23301650.v1 (accessed on 1 September 2022).

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
