# Peer review of "Keep Your Eggs Away: Ant Presence Reduces Ceratitis capitata Oviposition Behaviour through Trait-Mediated Indirect Interactions"

_insects, 2023, doi:10.3390/insects14060532_

Round 1

Reviewer 1 Report

In this manuscript, Smargiassi et.al., showed that females of C. capitata avoided the plum fruit that was exposed to Mediterranean ants, Crematogaster scutellaris and Tapinoma nigerrimum. This aversive behavioral response is most likely due to the chemicals deposited by ants on the fruit surface. This is an interesting observation. I have a few minor comments and I request the authors to discuss this in the discussion.

1.    In Figure 2, it seems that no significant aversive response (as shown by total time spent) was obtained on Day 3. This is true for both Crematogaster scutellaris and Tapinoma nigerrimum. Is this because of some habituation effect? The effect of exposure was pre-dominant only on Day 1.  Prolonged exposure did not increase the aversive response.

2.    The graph in Figure 2 looks different when compared to C scutellaris with T nigerrimum. Does it say something about the nature of the chemical cues emitted by two different Mediterranean ants?

3.    I am sorry if I missed the information. Could you kindly say whether the binary choice assay has been carried out in dark? This will tell us that the aversive behavioral response is driven by olfactory cues.

4.    Have you or anyone tried to wash the fruit with hexane or alcohol. If the chemical cues on the surface of the fruit drive the behavioral response, then one would expect that washed fruit has lesser effect on fly’s behavior.

Author Response

COMMENT: In Figure 2, it seems that no significant aversive response (as shown by total time spent) was obtained on Day 3. This is true for both Crematogaster scutellaris and Tapinoma nigerrimum. Is this because of some habituation effect? The effect of exposure was pre-dominant only on Day 1.  Prolonged exposure did not increase the aversive response.

REPLY: Interesting comment. We argued that it is due to the evaporation of the pheromone but yes, it is possible, although this species appears not to be sensible to habituation. We added a brief sentence about this.

COMMENT:  The graph in Figure 2 looks different when compared to C scutellaris with T nigerrimum. Does it say something about the nature of the chemical cues emitted by two different Mediterranean ants?

REPLY: Yes, this is possible, but in this phase we neither clearly know the nature of the compounds involved in this process, thus a precise assessment of the chemistry is one of the next programmed steps.

COMMENT:  I am sorry if I missed the information. Could you kindly say whether the binary choice assay has been carried out in dark? This will tell us that the aversive behavioral response is driven by olfactory cues.

REPLY: Good comment. We performed tests under natural daylight or even under lit neon lamps. However, we randomly changed the position of the treated fruits in each test, thus the visual cues should not have an effect.

COMMENT: Have you or anyone tried to wash the fruit with hexane or alcohol. If the chemical cues on the surface of the fruit drive the behavioral response, then one would expect that washed fruit has lesser effect on fly’s behavior.

REPLY: No we didn’t, and to our knowledge in literature no one else did. But it actually might be an interesting upgrade of the experimental setup.

Reviewer 2 Report

Dear authors,

I have read the manuscript “Keep your eggs away: ant presence reduces Ceratitis capitata oviposition behaviour through trait-mediated indirect interactions ”. The authors tested the effect of TMIIs of two Mediterranean ant species, Crematogaster scutellaris and Tapinoma nigerrimum, on the oviposition behaviour of Ceratitis capitata (Diptera: Tephritidae), one of the world's most economically damaging pests attacking fruits. Results of both ant’s species tests showed a significantly shorter time spent by ovipositing medflies on ant-exposed plums and a lower number of pupae, if compared to the control group.

This study contributes to the understanding of indirect ant-pest interactions in Mediterranean agricultural settings and points out the potential of utilising ant-borne semiochemicals in sustainable IPM strategies. The note is well written. The topic is interesting and new. The manuscript is very interesting.

I did have a two points that require clarification:

Line 96: Please add report examples.

Lines 125-128: I suggest changing the last sentence of introduction into a hypothesis.

Author Response

COMMENT: Line 96: Please add report examples.

REPLY: Done.

COMMENT: Lines 125-128: I suggest changing the last sentence of introduction into a hypothesis.

REPLY: Done.

Reviewer 3 Report

Title: Keep your eggs away: ant presence reduces Ceratitis capitata oviposition behaviour through trait-mediated indirect interactions

This paper aims to test whether chemicals of two common ant species reduce oviposition of Ceratitis capitata, a serious fly pest. Hypothesis, study design, and results are straightforward, and introduction and discussion are well written. This paper may give a new opportunity for biological control for the target pest. I recommend this paper for publishing in Insects.

However, please check the following wording errors. Reference spacing

Line 55: sustainable[2–4] -à sustainable [2–4], line 112: ecosystems[35–37] -à ecosystems [35–37] There are many same errors throughout the paper.

Line 138: Benelli et al., [45]. -à Benelli et al. [45].

Line 139: (relative humidity= 80 %), -à (relative humidity = 80%),

Author Response

We are sorry for the text typos, we accurately checked the manuscript, and we hope now there are no other mistakes.